# Comparing Styles across Languages

**Shreya Havaldar, Matthew Pressimone, Eric Wong, & Lyle Ungar**

University of Pennsylvania

{shreyah,mpressi,exwong,ungar}@seas.upenn.edu

## Abstract

Understanding how styles differ across languages is advantageous for training both humans and computers to generate culturally appropriate text. We introduce an explanation framework to extract stylistic differences from multilingual LMs and compare styles across languages. Our framework (1) generates comprehensive style lexica in any language and (2) consolidates feature importances from LMs into comparable lexical categories. We apply this framework to compare politeness, creating the first holistic multilingual politeness dataset and exploring how politeness varies across four languages. Our approach enables an effective evaluation of how distinct linguistic categories contribute to stylistic variations and provides interpretable insights into how people communicate differently around the world.

## 1 Introduction

Communication practices vary across cultures. Inherent differences in how people think and behave (Lehman et al., 2004) influence cultural norms, which in turn, have a significant impact on communication (Moorjani and Field, 1988). One key way that cultural variation influences communication is through *linguistic style*. In this work, we introduce an explanation framework to extract stylistic differences from multilingual language models (LMs), enabling cross-cultural style comparison.

Style is a complex and nuanced construction. In communication, style is heavily used to convey certain personal or social goals depending on the speakers' culture (Kang and Hovy, 2021). For example, cultures that are *high-context*[1] tend to use more indirect language than those that are *low-context* (Kim et al., 1998), and cultures that have a high *power-distance*[2] tend to have more formal

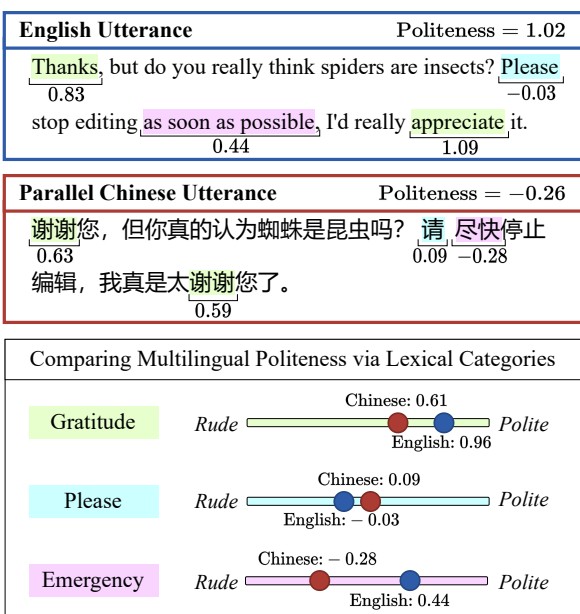

Figure 1: Explaining how politeness differs between parallel sentences in English and Chinese. Though these two sentences are identical in *content*, they differ in *style*. Our framework provides a way to quantitatively compare politeness in English and Chinese by comparing the importance of interpretable lexical categories.

interactions in a workplace setting. (Khatri, 2009).

For example, consider the two conversation snippets in Figure 1. Though these two utterances are direct translations of each other, a multilingual LM fine-tuned for politeness classification outputs different labels, with the Chinese utterance labeled as impolite. We ask a bilingual English/Chinese speaker to provide further insights, and she observes that the same request to "stop editing as soon as possible" sounds more aggressive (and thus, impolite) in Chinese than in English, as cultural norms in China do not typically condone giving such harsh, direct instructions to a stranger.

Stylistically appropriate language is crucial to successful communication within a certain culture. However, multilingual LMs sometimes struggle to

---

[1]Conversations in *high-context* cultures have a lot of subtlety and require more collective understanding. (Hall, 1976)

[2]*Power distance* refers to the strength of a society's social hierarchy. (Hofstede, 2005)

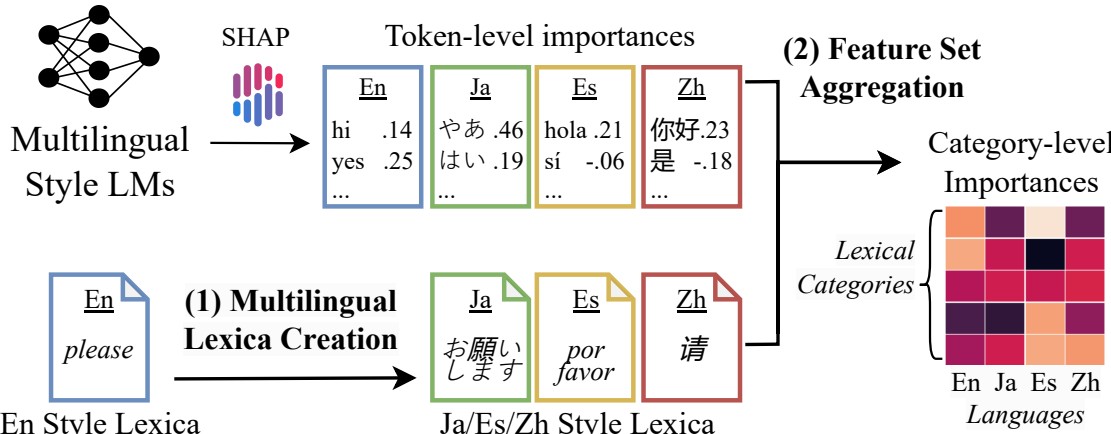

Figure 2: Our two-part explanation framework. Note that we use standard ISO language codes: En, Ja, Es, and Zh for English, Japanese, Spanish, and Chinese respectively.

generate language that is stylistically appropriate in non-English languages (Hershcovich et al., 2022; Ersoy et al., 2023; Zhang et al., 2022). Standard training methods for multilingual models lead to little stylistic variation in generated text across languages (Pires et al., 2019; Libovický et al., 2020; Muller et al., 2021), and multilingual systems rarely address these socially-driven factors of language (Hovy and Yang, 2021). As a result, downstream applications of these systems, like chatbots, are not as usable or beneficial to a non-American audience (Bawa et al., 2020; Havaldar et al., 2023a).

One step towards correcting this is to understand *how styles differ across languages*. Though modern multilingual LMs struggle with *generating* stylistically appropriate language, they are generally quite successful at *classifying* stylistic language (Briakou et al., 2021; Plaza-del Arco et al., 2020; El-Alami et al., 2022; Srinivasan and Choi, 2022). Psychological or linguistic studies to analyze language are resource-heavy and time-intensive; alternatively, we can utilize these trained LMs to computationally capture both the overt and subtle ways a language reflects a certain style.

Most current methods to extract feature importances from multilingual LMs are at the token-level (Lundberg and Lee, 2017; Ribeiro et al., 2016; Sundararajan et al., 2017) and specific to each language; as a result, there is no "common language" for comparison. In addition, most explanations are not easily human-interpretable, and it is difficult to extract useful takeaways from them.

In this work, we present a framework to extract differences in style from multilingual LMs and explain these differences in a functional, interpretable way. Our framework consists of two components:

1. **Multilingual Lexica Creation:** We utilize embedding-based methods to translate and expand style lexica in any language.

2. **Feature Set Aggregation:** We extract feature importances from LMs and consolidate them into comparable lexical categories.

To study how styles differ across languages, we create a holistic politeness dataset that encompasses a rich set of linguistic variations in four languages. Politeness varies greatly across languages due to cultural influences, and it is necessary to understand this variation in order to build multilingual systems with culturally-appropriate politeness. Trustworthy and successful conversational agents for therapy, teaching, customer service, etc. must be able to adapt levels and expressions of politeness to properly reflect the cultural norms of a user.

Previous work by Danescu-Niculescu-Mizil et al. (2013) and Srinivasan and Choi (2022) uses NLP to study politeness, but their datasets reflect only a small subset of language — the conversational utterances they analyze consist of only questions on either extremes of the impolite/polite spectrum.

Our dataset is the first *holistic* multilingual politeness dataset that includes *all types of sentences* (i.e. dialogue acts) across the *full impolite/polite spectrum*. We include English, Spanish, Japanese, and Chinese — we select these languages as they are all high-resource, (and therefore well-supported by modern LMs) and each have a unique way of expressing politeness. For instance, politeness in Japan frequently arises from acknowledging the place of others (Spencer-Oatey and Kádár, 2016),

while politeness in Spanish-speaking countries often relies on expressing mutual respect (Placencia and Garcia-Fernandez, 2017). This global variation in politeness (Leech, 2007; Pishghadam and Navari, 2012) makes it an important style to understand for effective cross-cultural communication.

Our contributions are as follows:

1. We present an explanation framework to extract differences in styles from multilingual LMs and meaningfully compare styles across languages.

2. We provide the first holistic politeness dataset that reflects a realistic distribution of conversational data across four languages.

3. We use this framework and dataset to show differences in how politeness is expressed (e.g. words like "bro" and "mate" are rude in English, but polite in Japanese, and yes/no questions are rude in Chinese but not in English.)

Figure 1 shows an example comparison of English and Chinese politeness using this framework. We explain differences in politeness using both lexical categories and dialogue acts, with Figure 1 highlighting three chosen lexical categories. We make all code and data available publicly.[3]

## 2 A Framework for Multilingual Style Comparison

In this section, we detail our two-part framework for multilingual style comparison. **(1) Multilingual Lexica Creation** takes a curated style lexica and uses embedding-based methods to refine lexica translation into another language, creating a set of parallel lexical categories. **(2) Feature Set Aggregation** maps extracted feature importances from trained LMs to these parallel lexical categories across languages. This framework helps us to interpret what multilingual LMs learn about style during training, and allows us to meaningfully compare how style is expressed across languages.

### 2.1 Multilingual Lexica Creation (MLC)

Lexica provide an interpretable grouping of words into meaningful categories. The traditional use of lexica in NLP relies on a simple bag-of-words representation, but allows humans to easily visualize which lexical categories appear in a dataset.

[3] https://github.com/shreyahavaldar/multilingual_politeness

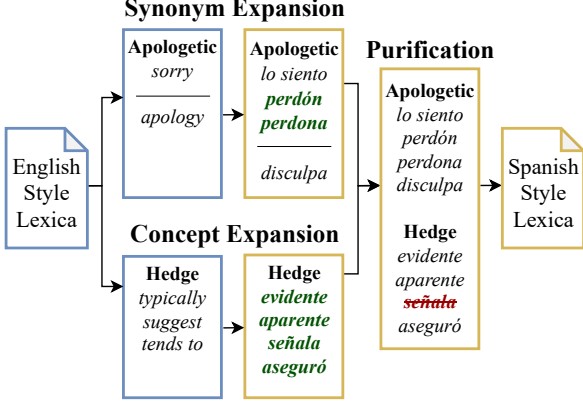

Figure 3: Multilingual Lexica Creation (MLC). We use word embeddings to expand and purify a style lexica translated from one language into another. This allows for maximum coverage per lexical category and corrects issues with standard 1:1 machine translation.

Much work has already been done in curating theory-grounded lexica that classify style. Danescu-Niculescu-Mizil et al. (2013) curate lexical strategies that inform politeness in English, and Li et al. (2020) extend these strategies to Chinese.

Though they provide insight into how politeness is expressed, lexica have limited predictive power; lexica-based models are drastically outperformed by modern LMs. Rather than relying on lexica to classify style, we instead use lexica to curate *a common language* for interpretable multilingual comparison. We present our method, Multilingual Lexica Creation (MLC), to expand a curated style lexica into multiple languages.

**Motivation.** When using standard 1:1 machine translation to translate a style lexica, a number of issues arise. Sometimes, there is no 1:1 mapping between words — a word in one language may have 0 words or 2+ words that express it in another. Additionally, context and culture influence how linguistic style is expressed (Moorjani and Field, 1988) — a word that reflects politeness in one language may not reflect politeness the same way in another. To combat these issues, MLC uses word embeddings to improve the translation process.

**Step 1: Expansion.** In the expansion step, we tackle the flawed assumption that there always exists a 1:1 mapping between words in different languages. First, we machine translate a curated style lexica into the target language. We refer to the words in this machine translated lexica as our set of *seed words*. Next, we embed each seed word us-

ing FastText (Bojanowski et al., 2017) and perform *synonym expansion* on each seed word and *concept expansion* on each lexical category. Figure 3 details this two-stage expansion process.

For synonym expansion, we find the nearest neighbors of each seed word in embedding space (within a tunable distance threshold) and append them to the corresponding lexical category. This adds any synonyms of seed words that may have been bypassed during machine translation. For instance, "sorry" in English is expanded to "lo siento", "perdón", and "perdona" in Spanish.

For concept expansion, we average the embeddings of all seed words within a lexical category and find the centroid embedding of the category. We then append the nearest neighbors of this centroid (within a tunable distance threshold) to the overall category. This adds any additional words conceptually similar to the lexical category that were not included via machine translation. For example, the Hedge category in Figure 3 is expanded to additionally include "evidente", "aparente", and "señala" in Spanish.

We choose FastText over other embedding models as it has a fixed vocabulary size, and thus, efficient nearest neighbors functionality. Additionally, FastText performs well in uncontextualized settings (Laville et al., 2020) and supports 157 languages. However, embeddings from any model can be used.

**Step 2: Purification.** In the purification step, we tackle the issue that a word reflecting a style in one language may not reflect that style in the same way when translated to another language. So, after combining the words returned from synonym and concept expansion, we ensure that each category of the expanded lexica contains words that are both *pertinent* and *internally correlated*.

We first filter out rare words (i.e. any words below a given usage frequency). This addresses issues where machine translation results in words not commonly used in day-to-day conversation.

Next, we ensure that the words in each lexical category reflect a given style similarly in the target language. Style is highly influenced by culture – a word that might indicate rudeness in English (e.g. "stubborn", "bossy", etc.) may not necessarily do so in other languages. To remove uncorrelated words, we first apply our lexica on any large corpus, and use a pre-trained LM to calculate a style score (e.g. politeness level) for each utterance in the corpus. Then, for each word $w$ within a lexical category $C$, we correlate the style scores of all utterances containing $w$ against the style scores of all utterances containing any word in $C$. We then remove all words that do not correlate positively with their category (product-moment correlation $< 0.15$). For example, "señala" does not have a similar role to other Hedge words when indicating politeness in Spanish ($r = -0.08$), and so, we remove it from the final lexica. This guarantees that all words within a lexical category play a similar role in determining an utterance's style score, ensuring internal correlation within each category.

MLC takes a curated style lexica and creates a parallel style lexica in a target language, correcting issues with 1:1 machine translation. Though the lexical categories are parallel, the words within each category are selected to best reflect how a style is expressed in the target language.

## 2.2 Feature Set Aggregation

We now seek to leverage the fact that trained multilingual LMs do successfully learn to encode how multiple languages reflect a certain style.

**Motivation.** However, traditional feature attribution methods to explain what LMs learn cannot be used for multilingual comparison, as extracted features always are specific to a single language. We bypass this limitation by aggregating extracted attributions into directly comparable categories.

Specifically, we extract feature attributions from trained multilingual style LMs and aggregate them into the parallel lexical categories from MLC. This enables an interpretable *common language* for comparison. We detail aggregation with token-level Shapley values (Lundberg and Lee, 2017), but any additive feature attribution method can be used.

**Token-to-word grouping.** Given a word $w$ consisting of tokens $w = [x_1, x_2, \ldots, x_K]$, let $[v_1, v_2, \ldots, v_K]$ be the corresponding token-level Shapley values. Equation (1) first aggregates token-level Shapley values into word-level importance scores.

$$\text{Imp}(w) = \sum_{k \, : \, x_k \in w} v_k \qquad (1)$$

**Category-level importances.** Next, we derive category-level importance scores for each lexical category by aggregating local word-level importance scores. For each category $C$, we iterate over all utterances and sum the word-level importance

scores for all words in $C$. Finally, we divide by $N$, or the total number of times a word in $C$ appears in the dataset. This process is detailed in Equation (2). Let $w_{ij}$ denote the $j$th word in the $i$th utterance.

$$\text{Imp}(C) = \frac{1}{N} \sum_{ij} \mathbb{1}_{[w_{ij} \in C]} \text{Imp}(w_{ij}) \quad (2)$$

This gives us an importance score for each lexical category across languages. Now, we can easily compare how important certain categories are at linguistically reflecting a style in different languages.

## 3 A Holistic Politeness Dataset

Politeness, like all linguistic styles, is a complex, nuanced construction. In order to compare how politeness differs across languages, it is necessary to analyze the full distribution of conversational data, without any oversimplifying assumptions. We follow the process of the Stanford Politeness Corpus (Danescu-Niculescu-Mizil et al., 2013) and TYDIP(Srinivasan and Choi, 2022) when creating and evaluating our politeness dataset, but with three key differences:

- We include *all dialogue acts* to replicate the real distribution of conversational data. (Both previous datasets only include questions.)

- We include *all annotated data* in model training and evaluation, as we want to compare language along the *full politeness spectrum*. (Both previous evaluations only consider the highest and lowest 25% of politeness utterances, eliminating all neutral utterances.)

- We treat politeness as a *regression task* rather than a binary classification task. (Both previous evaluations classify an utterance as "polite" or "impolite", destroying the nuance between slight and strong politeness.)

### 3.1 Data Collection & Annotation Overview

We provide a high-level overview of our dataset construction and annotation process, and give specific details in Appendix A.

**Dataset.** Our dataset contains 22,800 conversation snippets, or utterances, scraped from Wikipedia Talk Pages[4] in English, Spanish, Chinese, and Japanese (5,700 utterances per language).

---

[4]Wikipedia Talk Pages are used by editors of the platform to communicate about proposed edits to articles. This is the same domain as the Stanford Politeness Corpus and TYDIP.

Each utterance is 2-3 sentences long, and randomly sampled from all 41,000 scraped talk pages in each language. Note that we only scrape talk pages of articles that exist in all four languages, ensuring a similar distribution of topics across our dataset.

**Annotation.** To label the dataset, we use Prolific to source annotators. Respondents are required to be native speakers of the language they annotate, as well as indicate that it is their primary spoken language day-to-day. We include attention and fluency checks to ensure our annotations are of high quality. Annotators use a 5-point scale for labeling: "Rude", "Slightly Rude", "Neutral", "Slightly Polite", and "Polite", with three annotators labeling each utterance. We observe an average annotator agreement (Fleiss' kappa) of 0.186 across languages.

Given the highly subjective nature of politeness, we expect to see a score in this range. Additionally, we convert the Stanford Politeness Corpus (Danescu-Niculescu-Mizil et al., 2013) to a 5-point scale and observe a Fleiss' kappa of 0.153, indicating our agreement aligns with past work.

A key distinction between our dataset and the Stanford Politeness Corpus/TYDIPis that we do *not* normalize the scores of each annotator to be centered at neutral. Levels of politeness vary culturally (Leech, 2007; Pishghadam and Navari, 2012); we therefore do not make any assumptions about the politeness level of an average utterance.

Figure A6 provides a visualization of score distributions in our final annotated dataset. We observe that the average utterance from English, Spanish, and Chinese is closest to neutral, while the average Japanese utterance is closer to slightly polite.

## 4 Comparing Politeness via PoliteLex

PoliteLex (Li et al., 2020) consists of curated lexica to measure politeness in English and Chinese, based on the twenty politeness strategies introduced by Danescu-Niculescu-Mizil et al. (2013). We use MLC to expand PoliteLex to cover all four of our languages.

Chinese PoliteLex has some additional lexical categories, such as taboo words and honorifics, to account for cultural and linguistic norms in Chinese that do not have an English equivalent. As politeness is expressed more similarly within Eastern and Western cultures than between them (Spencer-Oatey and Kádár, 2016), we use English PoliteLex as the seed for Spanish and Chinese PoliteLex as

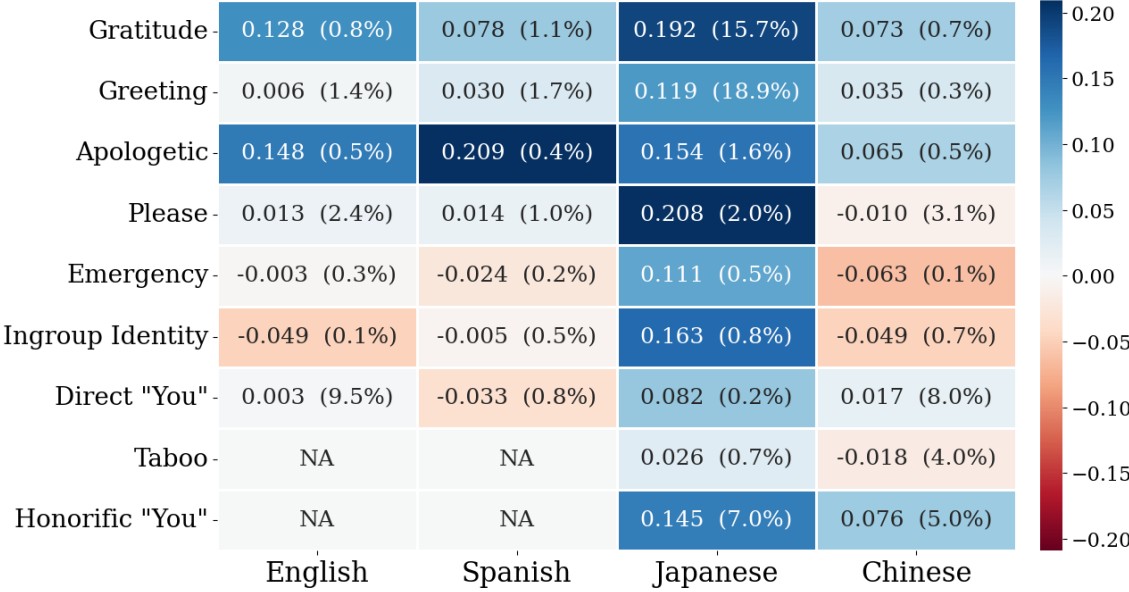

**PoliteLex: Average Category Importance & Category Frequency**

| | English | Spanish | Japanese | Chinese |
|---|---|---|---|---|
| Gratitude | 0.128 (0.8%) | 0.078 (1.1%) | 0.192 (15.7%) | 0.073 (0.7%) |
| Greeting | 0.006 (1.4%) | 0.030 (1.7%) | 0.119 (18.9%) | 0.035 (0.3%) |
| Apologetic | 0.148 (0.5%) | 0.209 (0.4%) | 0.154 (1.6%) | 0.065 (0.5%) |
| Please | 0.013 (2.4%) | 0.014 (1.0%) | 0.208 (2.0%) | -0.010 (3.1%) |
| Emergency | -0.003 (0.3%) | -0.024 (0.2%) | 0.111 (0.5%) | -0.063 (0.1%) |
| Ingroup Identity | -0.049 (0.1%) | -0.005 (0.5%) | 0.163 (0.8%) | -0.049 (0.7%) |
| Direct "You" | 0.003 (9.5%) | -0.033 (0.8%) | 0.082 (0.2%) | 0.017 (8.0%) |
| Taboo | NA | NA | 0.026 (0.7%) | -0.018 (4.0%) |
| Honorific "You" | NA | NA | 0.145 (7.0%) | 0.076 (5.0%) |

Figure 4: PoliteLex category-level importances across languages. Each importance score indicates the category's average numerical contribution to an utterance's politeness label, where $-2 =$ Rude, $0 =$ Neutral, and $2 =$ Polite. We additionally show the frequency of each category (% of total sentences that contain a word from the category.)

the seed for Japanese, to create a set of four lexica with parallel, comparable categories.

When purifying our expanded lexica in Spanish and Japanese, we use the full set of 41,000 scraped talk pages to calculate internal correlation and remove uncorrelated words from each category.

Next, we fine-tune XLM-RoBERTa models (Conneau et al., 2020) on our holistic politeness dataset (see Appendix A.4 for training details) and use the SHAP library (Lundberg and Lee, 2017) to extract Shapley values for each utterance. Finally, we apply Feature Set Aggregation to calculate importance scores for each PoliteLex category.

**Dataset coverage.** Table 1 analyzes our generated politeness lexica in Spanish and Japanese. We measure *dataset coverage* for each language – an utterance is "covered" if it contains at least one word in a lexical category, and we define dataset coverage as the percent of covered utterances. In both cases, the lexica generated by MLC has better coverage than 1:1 machine translation using Google Translate.

**Results.** Figure 4 details the resulting category-level importances from select PoliteLex categories, with the frequency of words in each category given in parentheses. Categories with positive importance are indicators of politeness, as the extracted

| Language | Lexica | % of Dataset Covered |
|---|---|---|
| English | PoliteLex | 98.0% |
| Chinese | PoliteLex | 83.7% |
| Spanish | MT | 94.4% |
| | **MLC** | **96.9%** |
| Japanese | MT | 57.0% |
| | **MLC** | **90.2%** |

Table 1: Lexical coverage of our holistic politeness dataset. We define *dataset coverage* as the percent of utterances containing a word in at least one lexical category. MLC produces lexica with better coverage than MT (machine translation).

Shapley values are highest for words within those categories. Similarly, categories with negative importance scores indicate rudeness.

For certain categories, we see a strong similarity across languages. Apologetic expressions (e.g. "I'm sorry", "my bad") and expressions of gratitude (e.g. "thank you", "I appreciate it") tend to universally indicate politeness. Interestingly, we see Japanese speakers using expressions of gratitude and apology with the highest frequency across languages.

We also notice interesting differences. The word

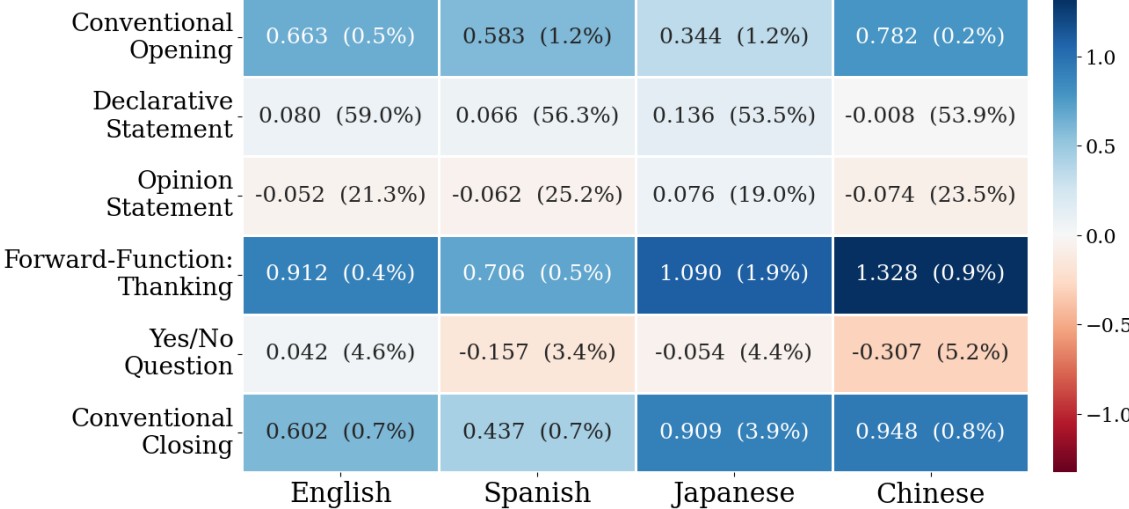

Figure 5: Dialogue act importances across languages. Each dialogue act importance score indicates the act's average numerical contribution to an utterance's politeness label, where $-2 =$ Rude, $0 =$ Neutral, and $2 =$ Polite. We additionally show the frequency of each dialogue act (% of total sentences classified as that act.)

"please" in English, Spanish, and Chinese does not indicate politeness, despite being used with similar frequency in all four languages. For example, the following English and Spanish utterances

"This has been debated to death; please read the archives."

"Antes de cuestionar si lo que digo es verdad, por favor trate de corroborarlo usted mismo." (*Before you question whether what I say is true, please try to verify it yourself.*")

are both labeled by annotators to be quite rude, despite containing the word "please". In Japanese however, "please" strongly indicates politeness.

Additionally, contrary to the findings of Li et al. (2020), expressions of in-group identity (e.g. "bro", "mate") are indicators of rudeness in English and Chinese, but indicators of politeness in Japanese. This may be because these terms are uncomfortably familiar, and so taken as rude, or due to sarcastic uses of these terms in English and Chinese. This phenomenon does not appear to be paralleled in Japanese, as terms of in-group identity are very polite. We give examples of top words in each PoliteLex category in Table A6 and our full set of results for all categories in Figure A7.

## 5 Comparing Politeness via Dialogue Acts

Given our dataset is the first politeness dataset to include multiple types of sentences (i.e. dialogue acts), we additionally apply the second part of our framework to dialogue act groupings as categories. In the previous section, we compared how *linguistic expressions* of politeness differ across languages. In this section, we seek to compare how the *linguistic form* of politeness differs as well.

To classify the dialogue acts of each utterance, we machine translate our dataset to English. We then run a trained English dialogue act classifier provided by Omitaomu et al. (2022) on the translated dataset and label each sentence of an utterance with one of 42 dialogue acts (Stolcke et al., 2000). Table 2 shows examples for select dialogue acts.

As dialogue acts are sentence-level, we modify Equation (1) to aggregate over all tokens in a given sentence, as opposed to all tokens in a given word. Finally, we treat each dialogue act as a unique category (analogous to a lexical category) and use our feature set aggregation method to map sentence-level SHAP values to their corresponding dialogue acts across our four languages.

**Results.** Figure 5 shows the average importance of each dialogue act, with the frequency of each dialogue act given in parentheses. Once again, we observe some similarities across languages: conventional openings, conventional closings, and sentences of thanks are strong indicators of politeness across languages.

However, statements appear to have differing roles across languages. Declarative statements

| Dialogue Act | Example |
|---|---|
| Conventional Opening | "Hi all, this article is in desperate need of attention." |
| Declarative Statement | "As it stands, we do not know the place or date of Shapiro's birth." |
| Opinion Statement | "There is no need to be combative and attack over typos." |
| Forward-Function: Thanking | "Looks good, thanks for the help!" |
| Yes/No Question | "Is the second picture an actual representation of the BCG vaccine?" |
| Conventional Closing | "Happy to discuss anything further." |

Table 2: Examples for selected dialogue acts from our holistic politeness dataset.

mostly lean polite across languages, while statements expressing an opinion lean slightly rude in English, Spanish, and Chinese. Surprisingly, yes/no questions only indicate politeness in English, and are viewed as mildly rude in all other languages, particularly Chinese. Consider the following yes/no questions in English and Chinese:

"To be pedantic, are we sure that he was born in Milton, West Dunbartonshire?"

"最后应用一节，有没有必要加入那么多图片？" ("*In the last application section, is it necessary to add so many pictures?*")

To an English speaker, both the English sentence and the Chinese translation appear to be similar levels of politeness. However, American annotators label the English question as "Neutral" while Chinese annotators label the Chinese question as "Slightly Rude," highlighting the ways in which cultural norms influence perceptions of politeness.

Interestingly, we do not observe any major differences in the frequency of dialogue acts across languages; conversations in all four languages appear to have a similar distribution of dialogue acts, though the average politeness of each dialogue act often varies based on language. Results for all dialogue acts are shown in Figure A8.

## 6 Ablation Analysis

The PoliteLex category-level importances in Figure 4 and dialogue act importances in Figure 5 are dependent on the Shapley values extracted from fine-tuned XLM-RoBERTa models. In this section, we analyze the effect of using alternate models and training paradigms.

**Effect of model size.** To investigate the role of LM size and architecture, we fine-tune Llama-2-7b (Touvron et al., 2023) to analyze politeness. Comparing the results from Llama-2-7b to the results

from XLM-RoBERTa, we notice stability in the *direction* of importance score (i.e. positive, negative, and neutral lexical categories/dialog acts are stable across both LMs). Interestingly, we observe differences in the *magnitude* of importance score (e.g. Llama-2-7b sees the "Greeting" category in Chinese to be a larger indicator of politeness than XLM-RoBERTa does).

**Effect of language-specific pretraining.** To investigate the role of language-specific pretraining, we fully fine-tune four RoBERTa models trained on only their respective languages. Similar to Llama-2-7b, we observe high similarity in the *direction* of the importance score compared to XLM-RoBERTa. We also notice much more similarity in the *magnitude* of importance scores. This may be due to inherent similarities between all RoBERTa models (parameter size, training methods, training data, etc.), which do not exist between XLM-RoBERTa and Llama-2-7b.

Our ablation analysis reveals that different language models pay attention to the same markers when learning to predict politeness, but learn to weigh these markers in different ways. Overall, we notice stability in which lexical categories and dialog acts indicate politeness vs. rudeness. This suggests that our groupings for feature set aggregation are both stable and successful. Section C contains additional details.

## 7 Related Work

**Multilingual style.** Previous work on multilingual style predominantly focuses on training LMs to perform cross-lingual and multilingual style classification and style transfer. Key styles studied include formality (Briakou et al., 2021; Krishna et al., 2022; Rippeth et al., 2022) and emotion (Öhman et al., 2018; Lamprinidis et al., 2021; Öhman et al., 2020), with another body of work focusing on style-

aware multilingual generation with any subset of chosen styles (Niu et al., 2018; Garcia et al., 2021).

**Explaining style.** One line of work builds on existing techniques (Lundberg and Lee, 2017; Ribeiro et al., 2016) to explain style within a single LM (Aubakirova and Bansal, 2016; Wang et al., 2021). Another line of work interprets style LMs by comparing learned features to those humans would consider important (Hayati et al., 2021), mapping feature attributions to topics (Havaldar et al., 2023b), and training models to output relevant features alongside their predictions (Hayati et al., 2023).

**Politeness.** Danescu-Niculescu-Mizil et al. (2013) presents one of the earliest quantitative analyses of linguistic politeness, with Srinivasan and Choi (2022) following in a multilingual setting. Other computational work focusing on politeness uses LMs to generate or modify text with a specified politeness level (Niu and Bansal, 2018; Fu et al., 2020; Mishra et al., 2022; Silva et al., 2022).

Previous work focused on multilingual style has little emphasis on investigating how style differs amongst languages. Additionally, most work on explaining style LMs is English-scoped, and thus, developed methods do not easily allow for cultural comparison. We are the first to present a method to compare styles across languages and provide quantitative, human-interpretable insights into how communication differs globally.

## 8 Conclusion

In this work, we present a framework to extract the knowledge implicit in trained multilingual LMs. This knowledge enables comparison of styles across languages via a human-interpretable common language for explanation. We also provide the first holistic multilingual politeness dataset, which we hope will encourage future research exploring cultural differences in style.

Our framework provides insights into how style is expressed differently across languages. These insights can improve multilingual LMs — understanding *how* and *why* an LM generation is not stylistically appropriate can inform culturally-adaptable models. These insights can also help people learning a second language become more aware of the language's culturally-informed stylistic nuances.

At a higher level, our framework provides a general methodology for explaining LMs; using feature attributions such as Shapley values to provide explanations in terms of human-interpretable categories (e.g. lexica and dialogue acts) gives explanations that are both grounded in the model and useful to humans.

## Limitations

When detailing our framework, we used FastText for Multilingual Lexica Generation and Partition SHAP (Lundberg and Lee, 2017) for Feature Set Aggregation. Our results are dependent on these two choices. Because FastText only supports word embeddings, we could only apply our MLC framework to words in the lexica. A contextual embedding could be used to additionally expand phrases. Additionally, we did not use native speakers to further purify our final lexica, as is the case in most past work curating multilingual lexica.

Individuals view politeness differently even within cultures, and as such, it is a highly subjective task. The subjectivity of politeness has been studied previously (Leech, 2007; Pishghadam and Navari, 2012; Spencer-Oatey and Kádár, 2016; Placencia and Garcia-Fernandez, 2017) and as a result, there is no "correct" label for an utterance; this subjectivity contributes heavily to our annotator agreement scores and fine-tuned LM accuracy.

We also draw conclusions about politeness in various cultures from a single context. We only study Wikipedia Talk Page data, which reflects politeness in workplace communication, but expressions of politeness are likely to differ in other settings: non-work conversations, communication between friends or family, social media, etc.

## Ethics Statement

When comparing styles across languages in this study, we treat language and culture as monolithic. However, we recognize different communities of people who speak the same language will use it differently. For example, politeness is likely expressed differently in Spain vs. Mexico, even though they are both Spanish-speaking countries.

In studying politeness, we recognize that it is highly contextual – calling a stranger "bro" can be perceived as an insult, while calling a close friend "bro" can be viewed as an expression of bonding and closeness. Age, gender, and other demographics also play an important role in perceived po-

liteness in a conversation. Politeness is a deeply interpersonal act, but it is decontextualized when studied computationally; NLP studies of politeness destroy key components of it.

We additionally recognize the implications, both good and bad, of working towards culturally-appropriate stylistic language generation. Our method can be used to inform more culturally attuned LMs – multilingual polite agents are important for beneficial uses like teaching and therapy, but these polite agents could potentially also be used for more effective manipulation or misinformation tactics.

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

## A Dataset Construction and Annotation: Additional Details

### A.1 Data Collection

We scrape Wikipedia Editor Talk Pages in English, Spanish, Japanese, and Chinese using Wikimedia[5] and keep the Talk Pages discussing articles that are present in all four languages. Next, we extract utterances, or two–three sentence snippets from a single user, such that the length of a single utterance does not exceed 512 tokens. We anonymize utterances to remove names, replacing mentions with "@user" and hyperlinks, replacing hyperlinks with "<url>". We then randomly sample 5,700 utterances from each language.

We select 5,700 as the size of the dataset for each language based on a power calculation to observe a discernible difference in politeness level between four languages. We use the results from a pre-trained English-only politeness classifier taken from Hayati et al. (2021) to get the parameters for the power calculation ($\alpha = 0.05, \beta = 0.2$.)

### A.2 Annotation

To annotate our dataset, we set up surveys hosted using Prolific. We source annotators who are native speakers of the target language and additionally indicate that the target language is their primary spoken language day-to-day. We have each annotator label 100 utterances on a five-point scale: "Rude", "Slightly Rude", "Neutral", "Slightly Polite", and "Polite". Every 25 utterances, we give an attention check to each annotator. The attention check is in the form 'To ensure you are still paying attention, please select [indicated option]" and translated to the target language. If the annotator fails any one of the four total attention checks, we discard all of their annotations. We additionally use this as a fluency check, as the attention check is fully written in the target language. We set up the survey such that each utterance is labeled by three different annotators.

Each annotator is paid $12 an hour, and each survey took, on average, 20 minutes to complete. We do not notice any differences in survey time between languages.

### A.3 Data Processing

Upon the completion of all four Prolific studies, we calculate annotator agreement using Fleiss' Kappa.

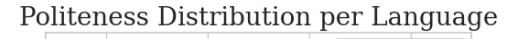

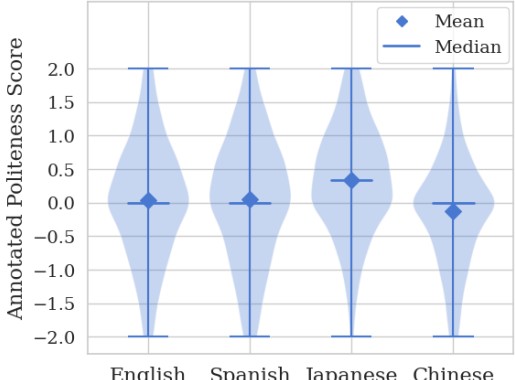

Figure A6: Distribution of annotated politeness scores per language.

| Language | Learning Rate | Test RMSE | Test $r$ |
|---|---|---|---|
| English | $1e-5$ | .667 | .662 |
| Spanish | $1e-5$ | .738 | .642 |
| Japanese | $1e-5$ | .659 | .650 |
| Chinese | $5e-5$ | .612 | .646 |

Table A3: Details on model training. We fine-tune four XLM-RoBERTa models on our holistic politeness dataset. We show Test RMSE (root mean squared error) and product-moment correlation $r$ for the final model.

Table A4 shows agreement for each language. We observe a slightly higher agreement in all languages than that of the Stanford Politeness Corpus (Fleiss' kappa = 0.15)(Danescu-Niculescu-Mizil et al., 2013), indicating the highly subjective nature of politeness.

Next, we convert the annotations to a numeric scale: "Rude"$= -2$, "Slightly Rude"$= -1$, ..., "Polite"$= 2$. Finally, we use each utterance's average numeric rating as its final label.

### A.4 Model Fine-Tuning

To create LMs capable of detecting politeness in multiple languages, we fine-tune RoBERTa models on our holistic politeness dataset.

We train four XLM-RoBERTa (Conneau et al., 2020) models, one for each language. For training, we randomly split each language's utterances into an 80/10/10 train/validation/test split. We use the Huggingface Trainer library to fine-tune our models. All our models are trained for 50 epochs, and we select the epoch with the lowest validation loss as our final model. The results for the best iteration of each model can be found in Table A3.

---

[5] https://dumps.wikimedia.org/backup-index.html

| Language | Num Utterances | Fleiss's Kappa | Mean Politeness | Standard Deviation |
|----------|----------------|----------------|-----------------|--------------------|
| English  | 5700           | 0.186          | 0.035           | 0.88               |
| Spanish  | 5700           | 0.188          | 0.053           | 0.93               |
| Japanese | 5700           | 0.162          | 0.332           | 0.82               |
| Chinese  | 5700           | 0.206          | -0.123          | 0.80               |

Table A4: Dataset Statistics: annotator agreement and details on annotation distribution.

Note that we choose to train separate models for each language (as opposed to a single one for all languages) as we want our trained models to encode how politeness is expressed in each target language uniquely, without being influenced by utterances from other languages.

## B Comparing Politeness Across Languages: Full Results

**PoliteLex.** Figure A7 shows category-level importance scores across all PoliteLex categories. Note that "NA" indicates that the category is not present in that language. We use English PoliteLex as the seed for Spanish PoliteLex, and Chinese PoliteLex as the seed for Japanese PoliteLex; the categories present in the expanded lexica parallel those present in the seed lexica.

Interesting findings include:

- Deference is seen as polite in English, Spanish, and Japanese, but is seen as slightly rude in Chinese.

- Contrary to the findings of Danescu-Niculescu-Mizil et al. (2013), we observe no concrete difference in politeness between utterances that use the subjunctive form of a word ("could", "would") vs. the indicative form ("can", "will").

- Indirectness ("by the way") is seen as polite in English and Japanese, but rude in Spanish and Chinese.

An overall analysis of our results reveals that all Japanese PoliteLex strategies indicate either neutrality or politeness. This is not the case in other languages; certain strategies do indicate rudeness. These findings suggest that politeness is expressed in a unique way in Japan, and perhaps rudeness is more subtle and cannot be fully captured by the lexical categories used in Chinese PoliteLex.

**Dialogue acts.** Figure A8 shows act-level importance scores across all dialogue acts, or types of sentences. We only show and compare dialogue acts that appear over ten times in each language, to ensure we have enough data for a meaningful comparison.

Interesting findings include:

- Rhetorical questions are seen as rude in English, Spanish, and Chinese, but closer to neutral in Japanese.

- "Wh-" questions (e.g. "Why did you do that?" or "Where are you going?") are seen as universally rude.

- Apology statements are seen as polite in all languages, but most polite in Chinese, despite the fact that Chinese speakers use apology statements with the lowest frequency.

Unlike PoliteLex categories, certain dialogue acts do reflect rudeness in Japanese, namely "Wh-" questions and statements that summarize or reformulate as a response.

## C Ablation Analysis: Additional Details

To fine-tune Llama-2-7b with the computational resources available to us, we use parameter efficient fine-tuning (Liu et al., 2022). Specifically, we use the PEFT library (Mangrulkar et al., 2022) to fine-tune a subset of the weights. We use a learning rate of $1e-5$ and default LORA parameters. We achieve similar performance to the metrics shown in Table A3.

To fine-tune the monolingual models (shown in Table A5), we use an identical set-up as Table A3, as the architecture of monolingual RoBERTa models and multilingual XLM-RoBERTa models is identical. Again, we observe similar performance to the metrics shown in Table A3.

We provide the resulting heatmaps in our repository: https://github.com/shreyahavaldar/multilingual_politeness

| Setting | Model Name |
|---|---|
| Monolingual English | roberta-base (Liu et al., 2019) |
| Monolingual Spanish | bertin-roberta-base-spanish (De la Rosa et al., 2022) |
| Monolingual Chinese | chinese-roberta-wwm-ext (Cui et al., 2020) |
| Monolingual Japanese | japanese-roberta-base (Cho and Sawada, 2021) |
| Multilingual | Llama-2-7b (Touvron et al., 2023) |

Table A5: LMs used in our ablation studies. We experiment with a different multilingual model to investigate the role of model size and architecture, as well as four separate monolingual models to investigate the role of language-specific pretraining.

| PoliteLex Category | Three Most Frequent Words (English) |
|---|---|
| Gratitude | thanks, thank you, i appreciate |
| Deference | good, great, interesting |
| Greeting | hello, hi, hey |
| Apologetic | sorry, apologize, apologies |
| Please | please, pls, plse |
| Please Start | please |
| Indirect (btw) | by the way, btw |
| Direct Question | what, when, how |
| Direct Start | but, and, so |
| Subjunctive | could you, would you |
| Indicative | can you, will you |
| 1st Person Start | i, my, mine |
| 1st Person Plural | we, us, our |
| 1st Person | i, my, me |
| 2nd Person | you, your, u |
| 2nd Person Start | you, your |
| Hedges | should, think, seems |
| Factuality | actually, really, in fact |
| Emergency | right now, immediately, at once |
| Ingroup Identity | mate, homie, dude |
| Praise | great, excellent, super |
| Promise | sure, must, certainly |
| Together | together |
| Direct "You" | you, u |
| Positive | like, well, good |
| Negative | issue, wrong, problem |

Table A6: For each PoliteLex category, we show the three most frequently occurring words in our holistic politeness dataset. Note that this table only shows words in English PoliteLex; Spanish/Chinese/Japanese PoliteLex contains different words in each category.

## PoliteLex: Average Category Importance & Category Frequency

| | English | Spanish | Japanese | Chinese |
|---|---|---|---|---|
| Gratitude | 0.128 (0.8%) | 0.078 (1.1%) | 0.192 (15.7%) | 0.073 (0.7%) |
| Deference | 0.040 (2.7%) | 0.024 (6.3%) | 0.147 (7.4%) | -0.032 (3.6%) |
| Greeting | 0.006 (1.4%) | 0.030 (1.7%) | 0.119 (18.9%) | 0.035 (0.3%) |
| Apologetic | 0.148 (0.5%) | 0.209 (0.4%) | 0.154 (1.6%) | 0.065 (0.5%) |
| Please | 0.013 (2.4%) | 0.014 (1.0%) | 0.208 (2.0%) | -0.010 (3.1%) |
| Please Start | -0.032 (1.5%) | -0.028 (0.4%) | NA | NA |
| Indirect (btw) | 0.053 (0.1%) | -0.028 (0.3%) | 0.094 (20.6%) | -0.054 (0.1%) |
| Direct Question | -0.006 (14.7%) | -0.040 (6.6%) | 0.091 (8.5%) | -0.049 (6.3%) |
| Direct Start | -0.005 (3.0%) | -0.020 (4.5%) | NA | NA |
| Subjunctive | 0.106 (0.1%) | 0.025 (1.5%) | NA | NA |
| Indicative | 0.114 (0.1%) | 0.000 (9.9%) | NA | NA |
| 1st Person Start | 0.005 (16.6%) | 0.002 (3.4%) | NA | NA |
| 1st Person Plural | 0.016 (5.2%) | -0.021 (1.4%) | 0.091 (0.0%) | -0.018 (0.6%) |
| 1st Person | 0.012 (41.0%) | -0.008 (16.6%) | 0.124 (8.8%) | -0.041 (23.5%) |
| 2nd Person | 0.005 (12.0%) | -0.003 (9.7%) | 0.066 (1.1%) | 0.017 (8.0%) |
| 2nd Person Start | -0.025 (1.5%) | -0.031 (0.3%) | NA | NA |
| Hedges | 0.003 (29.2%) | 0.012 (41.5%) | 0.098 (45.4%) | -0.026 (48.4%) |
| Factuality | -0.036 (3.8%) | -0.000 (7.4%) | 0.080 (0.7%) | -0.013 (0.6%) |
| Emergency | -0.003 (0.3%) | -0.024 (0.2%) | 0.111 (0.5%) | -0.063 (0.1%) |
| Ingroup Identity | -0.049 (0.1%) | -0.005 (0.5%) | 0.163 (0.8%) | -0.049 (0.7%) |
| Praise | 0.016 (0.8%) | 0.011 (2.5%) | 0.139 (9.5%) | -0.032 (3.6%) |
| Promise | -0.002 (2.6%) | 0.013 (5.5%) | 0.134 (2.7%) | -0.026 (3.8%) |
| Together | 0.034 (0.3%) | -0.005 (0.1%) | 0.077 (0.1%) | -0.043 (0.3%) |
| Direct "You" | 0.003 (9.5%) | -0.033 (0.8%) | 0.082 (0.2%) | 0.017 (8.0%) |
| Positive | 0.017 (39.8%) | 0.011 (39.4%) | NA | NA |
| Negative | -0.036 (34.6%) | -0.033 (39.6%) | NA | NA |
| Taboo | NA | NA | 0.026 (0.7%) | -0.018 (4.0%) |
| Honorific "You" | NA | NA | 0.145 (7.0%) | 0.076 (5.0%) |

Figure A7: The full set of PoliteLex category-level importances across languages. Each importance score indicates the category's average numerical contribution to an utterance's politeness label, where $-2 = $ Rude, $0 = $ Neutral, and $2 = $ Polite. We additionally show the frequency of each category (% of total sentences that contain a word from the category.)

# Dialogue Acts: Average Act Importance & Act Frequency

| | English | Spanish | Japanese | Chinese |
|---|---|---|---|---|
| Declarative Statement | 0.080 (59.0%) | 0.066 (56.3%) | 0.136 (53.5%) | -0.008 (53.9%) |
| Yes/No Question | 0.042 (4.6%) | -0.157 (3.4%) | -0.054 (4.4%) | -0.307 (5.2%) |
| Conventional Closing | 0.602 (0.7%) | 0.437 (0.7%) | 0.909 (3.9%) | 0.948 (0.8%) |
| Acceptance | 0.206 (2.0%) | 0.199 (1.7%) | 0.227 (2.3%) | 0.149 (0.9%) |
| Quoted material | -0.021 (0.7%) | -0.044 (1.2%) | 0.059 (2.0%) | -0.107 (3.3%) |
| Directive Action | 0.095 (2.1%) | -0.043 (1.2%) | 0.204 (2.2%) | -0.016 (2.2%) |
| "Wh-" Question | -0.334 (1.8%) | -0.398 (1.7%) | -0.138 (1.3%) | -0.441 (2.1%) |
| Opinion Statement | -0.052 (21.3%) | -0.062 (25.2%) | 0.076 (19.0%) | -0.074 (23.5%) |
| Open Question | -0.309 (0.5%) | -0.344 (0.5%) | 0.163 (0.6%) | -0.408 (0.7%) |
| Declarative Yes/No Question | -0.121 (0.8%) | -0.158 (0.6%) | -0.048 (0.4%) | -0.267 (0.6%) |
| Conventional Opening | 0.663 (0.5%) | 0.583 (1.2%) | 0.344 (1.2%) | 0.782 (0.2%) |
| Forward-Function: Apology | 0.445 (0.4%) | 0.285 (0.3%) | 0.800 (1.9%) | 1.068 (0.3%) |
| Completion | -0.093 (0.2%) | -0.146 (0.2%) | 0.004 (0.5%) | -0.040 (0.3%) |
| Forward-Function: Thanking | 0.912 (0.4%) | 0.706 (0.5%) | 1.090 (1.9%) | 1.328 (0.9%) |
| Rhetorical Question | -0.315 (0.6%) | -0.375 (0.5%) | -0.065 (0.3%) | -0.514 (0.8%) |
| Summarize/ Reformulate | -0.236 (0.3%) | -0.091 (0.3%) | -0.106 (0.4%) | 0.023 (0.4%) |

Figure A8: The full set of dialogue act importances across languages. Each dialogue act importance score indicates the acts' average numerical contribution to an utterance's politeness label, where $-2 = \text{Rude}$, $0 = \text{Neutral}$, and $2 = \text{Polite}$. We additionally show the frequency of each dialogue act (% of total sentences classified as that act.)