# OpenReview forum: "Comparing Styles across Languages"
_EMNLP/2023/Conference — EMNLP 2023 Main_

### Official Review · Reviewer_a7sm · 2023-08-04

**Soundness:** 4

**Excitement:**

4: Strong: This paper deepens the understanding of some phenomenon or lowers the barriers to an existing research direction.

**Paper Topic And Main Contributions:**

The authors present a framework to compare politeness knowledge included in LMs in order to compare how politeness style differ between four different languages: English, Spanish, Chinese and Japanese. This not only serves as a comparative linguistic study, but contribute in understanding how LMs handle these issues in different languages and if their style is appropiate for languages other than English. They also contribute with a multilingual politeness dataset.

Overall, this work provides a detailed study of politeness in multilingual LLMs and existing differences between languages in use of politeness that contribute to increase the explainability of LLM. Comparing the results on more LLMs would've been interesting and not really costly.

**Reasons To Accept:**

1. Well motivated.
2. Clear writing.
3. Release of code and data.
4. Provides a detailed insight on LLMs explainability on different languages.

**Reasons To Reject:**

1. Experiments could be more extensive.

**Reproducibility:**

5: Could easily reproduce the results.

**Reviewer Confidence:**

2: Willing to defend my evaluation, but it is fairly likely that I missed some details, didn't understand some central points, or can't be sure about the novelty of the work.

**Typos Grammar Style And Presentation Improvements:**

- Figures 4 and 5 feel too big.

---

> ### Author Rebuttal · Authors · 2023-08-28
>
> Thank you so much for your valuable feedback and for your excitement in our work! We look forward to improving our paper based on your comments:
>
> &nbsp;
>
> ### Using additional LMs
> Given our chosen LMs must be multilingual and open-source*, we additionally experiment on a series of language-specific RoBERTa models and LLAMA-2-7b. We report an interesting subset of our results below, as we are unable to link heatmaps as per the EMNLP rebuttal guidelines.
>
> **Getting SHAP values for closed-sourced models requires a significant compute cost that is out of scope of our paper. Running our same set of feature extraction experiments on GPT-4, for example, results in a proposed cost of over \\$34,900, hence we only use open-source LMs.*
>
> **Results for LLAMA-2-7b**
> To investigate the role of LM architecture, we fine-tune Llama-2-7b using PEFT (Mangrulkar et al. 2022) and report categories of interest.
>
> **Takeaways:** Compared to RoBERTA-XLM, we notice stability in *direction* of importance score (i.e. positive/negative/neutral lexical categories and dialog acts are stable across both LMs). Interestingly, we notice differences in *magnitude* (e.g. Llama-2-7b sees the "Greeting" category in Chinese to be a larger indicator of politeness than RoBERTa-XLM does).
>
> | Language | PoliteLex Category: *Greeting* | PoliteLex Category: *Emergency* | Dialog Act: *Conventional Opening* | Dialog Act: *Opinion Statement* |
> |--|--|--|--|--|
> | English |0.068|0.066|0.36|-0.056|
> | Spanish |0.001|0.02|0.765|0.087|
> | Japanese |0.189|0.089|0.612|0.033|
> | Chinese |0.173|0.148|0.519|-0.001|
>
> **Results for monolingual RoBERTa models**
>
> To investigate the role of language-specific pretraining, we fully fine-tune four separate RoBERTa models that have been trained on ONLY their respective languages.
>
> **Takeaways:** Similar to Llama-2-7b, we observe a very high similarity in *direction* of importance score compared to RoBERTa-XLM. We also notice more similarity in *magnitude* of importance scores. This may be due to inherent similarities between all RoBERTa models (parameter size, training methods, etc.), which do not exist between RoBERTa-XLM and Llama-2-7b.
>
> | RoBERTa Model | PoliteLex Category: *Greeting* | PoliteLex Category: *Emergency* | Dialog Act: *Conventional Opening* | Dialog Act: *Opinion Statement* |
> |--|--|--|--|--|
> | English (Liu et al., 2019) |0.012|-0.032|0.741|-0.083|
> | Spanish (De la Rosa et al., 2022) |0.011|0.02|0.626|-0.057|
> | Japanese (Cho and Sawada, 2021) |0.055|0.051|0.450|0.150|
> | Chinese (Cui et al., 2020) |0.053|0.016|0.265|-0.039|
>
> We have added an "LM Ablation Study" section to our appendix, where we will discuss the differences in results from different language models. Due to time constraints, these results use SHAP values from 20% of our dataset; we will include the full results in the camera-ready paper.
>
>
> &nbsp;
>
> ### Figure Size
> Based on your suggestion, we have shrunk Figures 4 and 5.
>
> &nbsp;
>
> Please let us know if you have any other questions or concerns, and we welcome additional feedback!

---

### Official Review · Reviewer_hCmg · 2023-08-05

**Soundness:** 4

**Excitement:**

4: Strong: This paper deepens the understanding of some phenomenon or lowers the barriers to an existing research direction.

**Paper Topic And Main Contributions:**

The paper presents a new method for generating reliable translations of lexicons categorized into meaningful groups. The approach involves expanding the lexicon with synonyms and conceptually related terms using word embeddings and nearest neighbor estimation. The expanded lexicon is then filtered based on style scores assigned by pretrained language models. The authors use this framework to study politeness across multiple languages, exploring the variations in the importance of lexical categories due to cultural differences. For studying politeness, they also create a corpus annotated with level of politeness on a 5 point scale.

**Reasons To Accept:**

- The suggested framework is simple and can be easily applied to other theoretically grounded style lexicons, provided that classifiers and datasets for comparison are available.
- Filtering the lexicon based on style ensures that the translations maintain linguistic coherence and stylistic consistency, making them more reliable to be used for downstream tasks as well.
- In the context of politeness the method provides an understanding of how the concept varies across languages, both at word level and utterance level.
- They create a relatively large corpus for politeness in 4 languages with open access.

**Reasons To Reject:**

- There is no discussion regarding the inherent bias and errors that can be introduced at different stages of the process, which could have propagated to the final analysis. For instance, the style scores are highly dependent on the annotated data which has a low agreement between annotators. Using these annotations to train the model could enhance any biases and uncertainties learned by the model.



**Reproducibility:**

4: Could mostly reproduce the results, but there may be some variation because of sample variance or minor variations in their interpretation of the protocol or method.

**Reviewer Confidence:**

3: Pretty sure, but there's a chance I missed something. Although I have a good feel for this area in general, I did not carefully check the paper's details, e.g., the math, experimental design, or novelty.

**Typos Grammar Style And Presentation Improvements:**

- Currently the introduction is a bit too long and lacks focus. It would be easier to follow the text if the introduction is made shorter focusing on the motivation, challenges and the contributions, giving a general overview rather than going into details.

---

> ### Author Rebuttal · Authors · 2023-08-28
>
> Thank you so much for your valuable feedback and for your excitement in our work! We look forward to improving our paper based on your comments:
>
> &nbsp;
>
> ### Discussion about potential bias
> We have added the following paragraph to the limitations section:
>
> **Bias and Error Analysis** Though our framework is general and can be applied to any LM + style lexicon, the specific results discussed in this work are dependent on many design choices. We use English and Chinese PoliteLex as our seed lexica, any knowledge gaps in these lexica may propagate to our expanded lexica too. We additionally train Roberta-XLM models on a human-annotated politeness dataset. Though we do our best to ensure sufficient annotation quality and use a standard training pipeline, any biases or uncertainties in the dataset or the LM may influence our downstream style comparisons. At a higher level, the results of our framework are highly specific to the dataset and LM chosen, and thus, any generalizations about politeness at the cultural level must be made with caution.
>
> &nbsp;
>
> ### Annotator agreement concerns
> To better evaluate our annotation quality, we calculate additional agreement metrics beyond Fleiss' kappa and compare them to widely used datasets in the style space. In addition to comparing against the Stanford Politeness Corpus, we also compare against the following:
>
> -  TyDIP (multilingual politeness) (Srinivasan and Choi 2022)
> -  EmoEvent (multilingual emotion) (Plaza del Arco et al. 2020)
> -  GoEmotions (English emotion) (Demszky et al. 2020)
> -  Formality Corpus (English formality) (Pavlick and Tetreault 2016)
> -  XFormal (multilingual formality) (Briakou et al. 2021)
>
> |Agreement Statistic|Range in Above Datasets|Ours|
> |--|--|--|
> |Intraclass Correlation|0.39-0.83|0.66|
> |Pearson Correlation|0.16-0.64|0.545|
> |Cohen's Kappa|0.09 - 0.55|0.22|
>
> *Overall, our dataset's annotator agreement is better than the average agreements of existing human-annotated datasets in the style space.* Note that the highest agreements are for emotions such as Love and Sadness, which are much clearer than politeness.
>
> Your comment raises an extremely important question -- How to improve annotation quality of subjective style tasks? However, given our dataset aligns with existing agreements, we feel answering this question is outside the scope of this work.
>
> &nbsp;
>
> ### Scoping introduction
> Based on your suggestions, we have scoped and shortened the introduction as follows:
> - We introduce our contributions + the purpose of this work earlier in the introduction
> - We make the discussion of Figure 1 shorter and more high-level, focusing on the overall motivation for our study
> - We remove repetitive sentences from consecutive paragraphs to better draw readers' attention to the key points
>
> &nbsp;
>
> Please let us know if you have any other questions or concerns, and we welcome additional feedback!

---

### Official Review · Reviewer_QT67 · 2023-08-08

**Soundness:** 3

**Excitement:**

4: Strong: This paper deepens the understanding of some phenomenon or lowers the barriers to an existing research direction.

**Paper Topic And Main Contributions:**

This paper studies how the expression of politeness is different across four languages, (i) English, (ii) Spanish, (iii) Japanese and (iv) Chinese. Authors propose a two-step approach, where they (i) Create multilingual lexica using existing multilingual embedding(MLM) and (ii) Do feature set aggregation to consolidate into appropriate lexical categories.  They extensively describe the motivation and steps they followed for creating multilingual lexica (Considering initial seed words, expanding to other languages by MLM, removing un-correlated words etc.), and feature set aggregation (getting token-level shapley values, getting category-level importance scores etc.), which clears overall set-up. They proposed a dataset (‘Holistic Politeness dataset’ (HPD hereafter)), which is similar to the Stanford politeness dataset and TYDIP, with differences like (i) HPD includes all dialogue acts, (ii) includes samples for borderline politeness annotations and (iii) considering the task as a regression one, unlike others where it is considered as a classification task. The data points are taken from Wikipedia Talk pages which I believe is a formal conversation dataset, and hence politeness variation in formal conversation can be easily studied.

To compare politeness, they used an existing curated lexica PoliteLex, which measures politeness using 20 politeness strategies in English and Chinese. They use the English one as a seed to construct the equivalent for Spanish and the Chinese one to construct the equivalent one for Japanese. So, they have four parallel lexica with comparable categories. They have finetuned XLM-RoBERTa models on the HPD and use the SHAP library to get shapley values. Feature set aggregation is applied to get categorical importance scores.

There have shown several interesting insights, (i) conversational openings, closings and thanking sentences are important indicators of politeness across all languages, (ii) Opinionated sentences are considered as slightly rude in English, Spanish and Chinese, (iii) Yes/No type questions are considered as mildly rude across all language except English, etc.


**Reasons To Accept:**

Interesting study. Politeness variation across languages is, in general, very useful. They can help us to fine-tune chatbot replies appropriately to different segments of society.

The methods presented in this paper are very simple yet effective. So, it was a pleasure for me to go through it.


**Reasons To Reject:**

The inter-annotator agreement score/ Fleiss’ kappa is 0.153, i.e. very low. Even though they have similar scores they got for the Stanford dataset, it raises serious questions on annotation quality. I understand that politeness is subjective and it depends on the individual. But in this case, we may not get very high score, but a moderate score is expected. Currently, it comes under ‘slight agreement’.


I urge authors to improve the coherence across sections by using a simple storyline. It will help the readers to understand it better.


I have reservations about the paper title. It is more generic, while the study done in this paper is more on the politeness dimension. So, I urge the authors to rephrase it.


I feel the contributions made in this paper are not enough for a long paper. I would recommend it for a short paper now.


**Reproducibility:**

4: Could mostly reproduce the results, but there may be some variation because of sample variance or minor variations in their interpretation of the protocol or method.

**Reviewer Confidence:**

3: Pretty sure, but there's a chance I missed something. Although I have a good feel for this area in general, I did not carefully check the paper's details, e.g., the math, experimental design, or novelty.

---

> ### Author Rebuttal · Authors · 2023-08-28
>
> Thank you so much for your valuable feedback and for your excitement in our work! We look forward to improving our paper based on your comments:
>
> &nbsp;
>
> ### Annotator agreement concerns
> To better evaluate our annotation quality, we calculate additional agreement metrics beyond Fleiss' kappa and compare them to widely used datasets in the style space. In addition to comparing against the Stanford Politeness Corpus, we also compare against the following:
>
> -  TyDIP (multilingual politeness) (Srinivasan and Choi 2022)
> -  EmoEvent (multilingual emotion) (Plaza del Arco et al. 2020)
> -  GoEmotions (English emotion) (Demszky et al. 2020)
> -  Formality Corpus (English formality) (Pavlick and Tetreault 2016)
> -  XFormal (multilingual formality) (Briakou et al. 2021)
>
>
> |Agreement Statistic|Range in Above Datasets|Ours|
> |--|--|--|
> |Intraclass Correlation|0.39-0.83|0.66|
> |Pearson Correlation|0.16-0.64|0.545|
> |Cohen's Kappa|0.09 - 0.55|0.22|
>
> *Overall, our dataset's annotator agreement is better than the average agreements of existing human-annotated datasets in the style space.* Additionally, the highest agreements are for emotions such as Love and Sadness, which are much clearer than politeness.
>
> Your comment raises an extremely important question -- How to improve annotation quality of subjective style tasks? However, given our dataset aligns with existing agreements, we feel answering this question is outside the scope of this work.
>
>  &nbsp;
>
> ### Improved coherence across sections
> Based on your suggestions, we have edited the paper as follows to have a better storyline across sections:
>
> - We emphasize the overall motivation at the start of every new section
> - We tie back to our research questions when discussing our results
>
> &nbsp;
>
> Please let us know if you have any other questions or concerns, and we welcome additional feedback!

---

### Meta-Review · Area_Chair_cMem · 2023-09-18

**Recommendation:** 4

**Metareview:**

The paper presents a new method for generating reliable translations of lexicons categorized into meaningful groups. The approach involves expanding the lexicon with synonyms and conceptually related terms using word embeddings and nearest neighbor estimation. The expanded lexicon is then filtered based on style scores assigned by pre-trained language models. The authors use this framework to study politeness across multiple languages, exploring the variations in the importance of lexical categories due to cultural differences. For studying politeness, they also create a corpus annotated with level of politeness on a 5 point scale.

Based on the reviews and the subsequent discussion during the discussion period, the significant aspects shared by reviewers include:

* The task is interesting and eventual applications (e.g. fine-tuning chatbots appropriately for different segments of society) is a reason to accept
* The paper is well motivated and generally well written. Issues that have been raised wrt coherence and missing discussion points have been addressed in the author response
* The presented framework is simple yet effective, which is a strong aspect of the paper
* A nice contribution to the community with the code and the multilingual politeness dataset

A shared issue brought up was that the experiments were not extensive enough to warrant a long paper. It should be noted that the author response addressed this well by providing new/additional analyses and results which should be included in the camera-ready, and make the paper stronger.

---

### Decision · Program_Chairs · 2023-10-07

**Decision:**

Accept-Main

**Comment:**

The paper presents a new method for generating reliable translations of lexicons categorized into meaningful groups. The approach involves expanding the lexicon with synonyms and conceptually related terms using word embeddings and nearest neighbor estimation. The expanded lexicon is then filtered based on style scores assigned by pre-trained language models. The authors use this framework to study politeness across multiple languages, exploring the variations in the importance of lexical categories due to cultural differences. For studying politeness, they also create a corpus annotated with level of politeness on a 5 point scale.

Based on the reviews and the subsequent discussion during the discussion period, the significant aspects shared by reviewers include:

* The task is interesting and eventual applications (e.g. fine-tuning chatbots appropriately for different segments of society) is a reason to accept
* The paper is well motivated and generally well written. Issues that have been raised wrt coherence and missing discussion points have been addressed in the author response
* The presented framework is simple yet effective, which is a strong aspect of the paper
* A nice contribution to the community with the code and the multilingual politeness dataset

A shared issue brought up was that the experiments were not extensive enough to warrant a long paper. It should be noted that the author response addressed this well by providing new/additional analyses and results which should be included in the camera-ready, and make the paper stronger.